# Superhydrophobic hemostatic nanofiber composites for fast clotting and minimal adhesion

Zhe Li [1,2], Athanasios Milionis [3], Yu Zheng [2], Marcus Yee[2], Lukas Codispoti[3], Freddie Tan [4],
Dimos Poulikakos [3]* & Choon Hwai Yap [2]*

Hemostatic materials are of great importance in medicine. However, their successful implementation is still challenging as it depends on two, often counteracting, attributes; achieving blood coagulation rapidly, before significant blood loss, and enabling subsequent facile wound-dressing removal, without clot tears and secondary bleeding. Here we illustrate an approach for achieving hemostasis, rationally targeting both attributes, via a super-hydrophobic surface with immobilized carbon nanofibers (CNFs). We find that CNFs promote quick fibrin growth and cause rapid clotting, and due to their superhydrophobic nature they severely limit blood wetting to prevent blood loss and drastically reduce bacteria attachment. Furthermore, minimal contact between the clot and the superhydrophobic CNF surface yields an unforced clot detachment after clot shrinkage. All these important attributes are verified in vitro and in vivo with rat experiments. Our work thereby demonstrates that this strategy for designing hemostatic patch materials has great potential.

[1] Guangdong Provincial Key Laboratory of Sensor Technology and Biomedical Instrument, School of Biomedical Engineering, Sun Yat-Sen University, Guangzhou 510006, China. [2] Department of Biomedical Engineering, National University of Singapore, Singapore 117583, Singapore. [3] Laboratory of Thermodynamics in Emerging Technologies, Department of Mechanical and Process Engineering, ETH Zurich, 8092 Zurich, Switzerland. [4] Department of Chemical and Biomolecular Engineering, National University of Singapore, Singapore 117585, Singapore. *email: dpoulikakos@ethz.ch; bieyapc@nus.edu.sg

Uncontrolled hemorrhage and wound infection are leading causes of death in the medical field of wound care[1–3]. Improperly dressed wounds will prolong healing time and impose a high infection risk[4], leading to significantly increased mortality and economic burden. To exemplify, $10 billion per year is spent on the treatment of complex wounds in North America[5], while the global wound care market is estimated to reach $22 billion by 2020[6]. Despite the progress in developing advanced hemostatic materials over the last few decades[1,3,7–9], there are still two major challenges to be addressed: excessive blood loss during the period that the clot is forming and strong clot adhesion on the hemostatic dressing that causes pain, secondary bleeding, and possible infection during the wound-dressing removal.

The conventional method to deal with bleeding is mechanically pressing the wound with a cotton gauze[10,11], which unavoidably absorbs blood and causes unnecessary blood loss and gauze adhesion onto the wound. Blood absorbed in the gauze forms a solid clot-gauze composite, forced peeling of which often tears the wound and causes secondary bleeding and pain. This makes it difficult to replace the old wound dressing without causing secondary infections or hemorrhage, in procedures ranging from common wounds to surgery, and to the extreme case of hemophilic patients[12], where excessive bleeding will occur before coagulation. To deal with these problems, active clotting agents (chitosan[3] or kaolin[7]) have been adopted into hemostatic materials, to reduce bleeding by expediting the coagulation process. However, such agents employ free micro-particles, which poses a safety threat of causing micro-thrombosis if they enter the vascular system[13,14]. Recently, researchers proposed using superhydrophobic (SHP) or superhydrophilic materials for hemostatic purposes. A superhydrophilic material (graphene sponge[8]) is reported to absorb water from the blood quickly, forming a dense layer of blood cells and platelets, thus promoting coagulation. Hydrophilic hemostatic material can also be prepared by spray coating β-chitosan on the porous nanofiber mat[15], and the hydrophilic β-chitosan coating can increase blood wettability and thus enhance clotting. Alternatively, a SHP coating can be applied on the back of the normal superhydrophilic gauze as an impervious layer to prevent blood loss through the gauze[9,16]. However, the core functionality of these approaches is still either based on a blood-absorbing hemostatic material (superhydrophilic) that does not minimize blood loss and secondary bleeding or a blood-repelling material (super-hemophobic) that simply repels blood but does not actively trigger clotting. Therefore, the aforementioned two key challenges on wound management still remain poorly addressed.

Here we report a strategy for achieving hemostasis by designing a SHP and blood-repelling surface that simultaneously achieves fast clotting with no blood loss, anti-bacterial property, and clot self-detachment. The non-wetting feature of the SHP hemostatic surface can withstand substantial blood pressure and help reduce blood loss and bacteria attachment. We find that carbon nanofibers (CNFs) immobilized on this surface can promote fast fibrin growth and thus clotting. Due to the presence of micro-air pockets within the blood–substrate contact area, there is minimal contact between the clot and the SHP CNF patch, leading to natural clot detachment after clot maturation and shrinkage, which reduces the peeling tension required to peel off the patch by about 1~2 orders of magnitude compared with a normal hydrophilic gauze or commercial hemostatic products. These features have been verified in vitro and in vivo, demonstrating the effectiveness of this strategy for designing hemostatic patch materials.

## Results

### Fibrin fiber formation on superhydrophobic CNF surfaces.
For effective hemostatic performance, our SHP surface is designed first

**Table 1 Superhydrophobic nanocomposite coatings.**

| Coating abbreviation | Substrate | Polymer matrix | CNF to polymer weight ratio |
| --- | --- | --- | --- |
| CNF/PTFE Ti | Ti plate | PTFE | 1:9 |
| CNF Ti mesh | Ti mesh # 60 | PTFE | 1:9 |
| CNF/PDMS Ti | Ti plate | PDMS | 1:2 |
| CNF gauze | Cotton gauze | PDMS | 1:2 |

to be strongly blood-repellent and second to be capable of triggering fast coagulation upon blood contact. Extreme blood repellency is achieved by spray coating of a nanocomposite dispersion, consisting of CNFs[17,18] (diameter: 100 nm, length: 20–200 μm), and polytetrafluoroethylene (PTFE)[19] or polydimethylsiloxane (PDMS)[20,21] onto a substrate (Table 1 and Supplementary Fig. 1). The generated surface has a dense layer of CNF network with micro/nano-roughness that is partially embedded in a hydrophobic polymer matrix (PTFE or PDMS) (Fig. 1a and Supplementary Fig. 1c). The use of hydrophobic base components, the micro/nano-scale topography from spray coating, and the morphology of the nanofibers collaboratively result in superhydrophobicity. The CNF/PTFE Ti surface had a water contact angle (WCA) of $162.1 \pm 2.9°$ (mean ± SD) and a water roll-off angle (WRA) of about 1°, and the CNF/PDMS Ti surface had a WCA of $154.9 \pm 0.6°$ and a WRA of about 4° (Fig. 1a and Supplementary Fig. 1a). The effect of CNF concentration on surface wetting was also investigated (Supplementary Fig. 1). Compared with water, blood has a smaller surface tension of 58 mN m$^{-1}$ (72 mN m$^{-1}$ for water)[22] and a complex composition, but our SHP CNF surfaces could still repel it, with blood contact angles (CAs) of $153.6 \pm 1.4°$ for the CNF/PTFE Ti surface and $151.4 \pm 1.8°$ for the CNF/PDMS Ti surface (Supplementary Fig. 1a).

Very interestingly and unexpectedly, we observed that, when blood or platelet poor plasma (PPP) with anticoagulant ethylenediaminetetraacetic acid (EDTA)[23] or sodium citrate[24] was brought into contact with our SHP CNF surfaces, long straight fibers (later confirmed to be fibrin) formed rapidly (Fig. 1b, c and Supplementary Movies 1–5). In the sliding test (Fig. 1b and Supplementary Fig. 2), abundant long fibers were generating at the receding side of the plasma droplet. These fibers pulled the droplet, retarding its sliding motion, until a critical angle was reached, causing catastrophic fiber fracture and allowing the droplet to roll off quickly (after the first or two fibers started to rupture, the remaining fibers could not hold the droplet weight and their rupturing occurred in the form of a rapid domino effect; Supplementary Movies 1 and 3). After the PPP droplet rolled off, visible fiber footprints were left on the surface (Fig. 1e), in the form of straight and ordered fibers on top of random CNFs (Supplementary Fig. 5e), aligning in the droplet rolling direction. The same was observed after blood droplet sliding (Supplementary Fig. 5d). Similar observations were made in a slightly different test, the touch-lift test, where a PPP or blood droplet was brought onto the CNF surface for a brief contact and then retracted (contact duration ~3 s; Fig. 1c), and fibrin fiber generation was also observed. Long and straight fibers (up to 300 μm) projecting outward from the contact center were detected under scanning electron microscopy (SEM; Supplementary Fig. 5f).

Fibrin fiber generation was observed despite the use of EDTA or sodium citrate (with anti-coagulation properties[23,24]), and occurred on both the SHP CNF/PTFE and CNF/PDMS surfaces (Fig. 1, Supplementary Figs. 3 and 4, and Supplementary Movies 1–5), but it was not observed on hydrophobic surfaces with a low CNF concentration or no CNF (Supplementary Fig. 6).

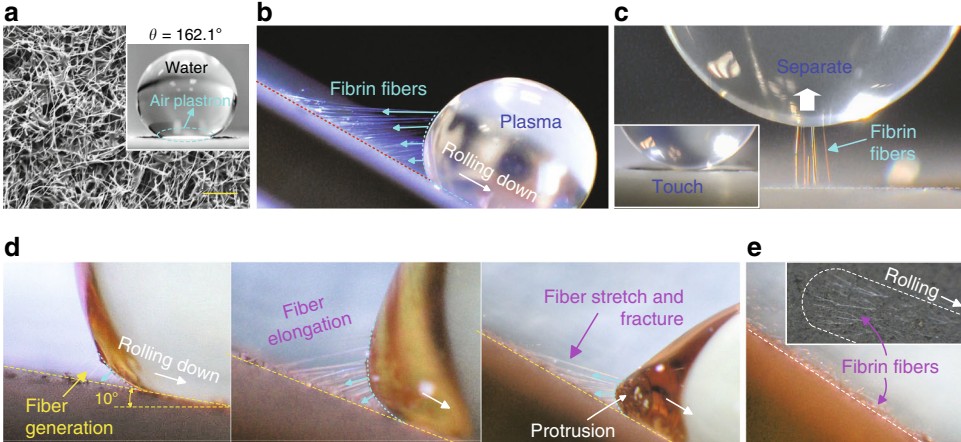

**Fig. 1 Fibrin fiber generation on the superhydrophobic CNF surface. a** SEM image of the superhydrophobic CNF/PTFE Ti surface and a water droplet sitting on this surface demonstrates excellent superhydrophobicity. **b** Long fibrin fibers generated at the receding side of a 10 µl EDTA PPP droplet. **c** Touch-lift test of a PPP droplet on the CNF surface, showing fiber generation upon EDTA PPP-substrate separation. **d** Zoomed-in sequential frames of the fiber generation and fracture during the EDTA PPP droplet (20 µl) rolling down motion (Supplementary Movie 1). **e** Visible fibrin fiber footprints left on the CNF surface after the PPP sliding test (inset is a top view). Scale bar is 10 µm in **a**.

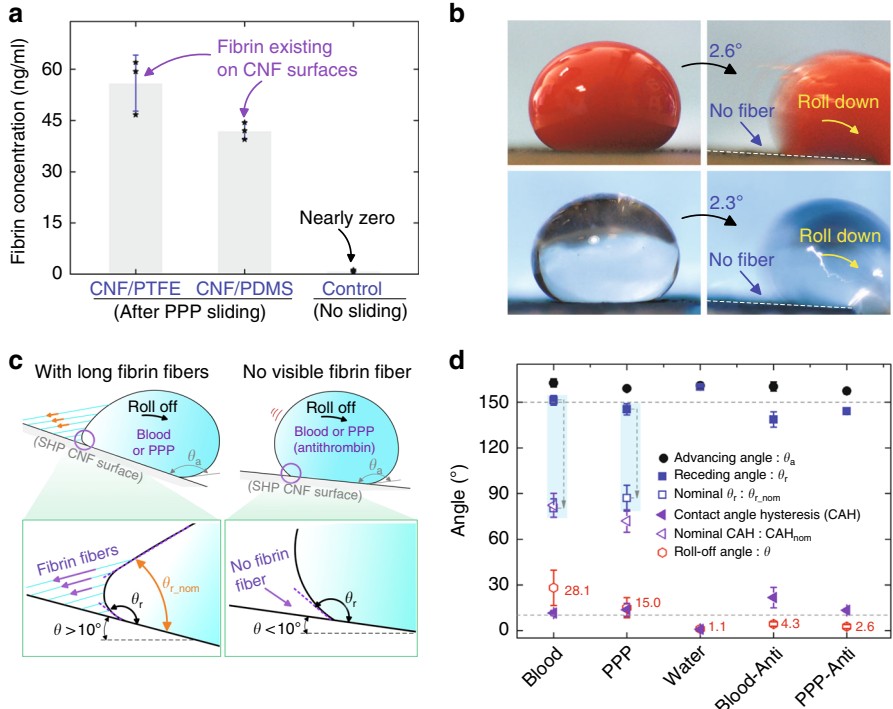

**Fig. 2 Fibrin confirmation. a** Fibrin ELISA test confirmed the existence of fibrin on the superhydrophobic CNF surfaces after a sliding test by plasma ($n = 3$). **b** Blood and PPP droplets with anti-thrombin (20 µl) quickly rolled down the superhydrophobic CNF/PTFE Ti surface at a small tilt angle without generating fibrin fibers at the receding side. **c** Actual and nominal receding angles for blood or PPP droplets sliding down the superhydrophobic CNF surface with or without fibrin fibers; $\theta_a$: the advancing angle, $\theta_r$: the actual receding angle, and $\theta_{r\_nom}$: the nominal receding angle; measurement of $\theta_{r\_nom}$ is explained in detail in Supplementary Fig. 8. **d** Dynamic contact angles for water droplets and blood or PPP droplets with and without anti-thrombin on the superhydrophobic CNF/PTFE Ti surface ($n = 4$); contact-angle hysteresis CAH $= \theta_a - \theta_r$, and nominal contact-angle hysteresis CAH$_{nom} = \theta_a - \theta_{r\_nom}$; droplet volume: 20 µl; Blood-Anti or PPP-Anti, respectively, represents blood or PPP with anti-thrombin. Data in **a** and **d** are shown as mean ± SD, the error bar represents SD, and individual data points in **a** are represented by black stars. Source data for **a** and **d** are provided as a Source Data file.

This suggested that the nano-engineered SHP CNF surface promoted fiber formation and the choice of PTFE or PDMS for CNF immobilization would not affect the fiber generation.

**Fibrin presence confirmation and generation mechanism.** To confirm that these fibers were indeed fibrin, we performed verification tests. Existence of fibrin, on both the SHP CNF/PTFE

and CNF/PDMS surfaces after PPP sliding, was verified through porcine fibrin enzyme-linked immunosorbent assay (ELISA) tests[25]. A standard dilute solution from the ELISA kit[25] was flushed over the CNF surface with and without PPP sliding to wash the generated fibrin fibers into the reaction well for a qualitative analysis. The positive results in Fig. 2a confirmed the fibers to be fibrin.

To further support this result, we found that adding anti-thrombin to the blood/plasma could prevent fiber formation (Supplementary Fig. 7a). As thrombin is a key factor for fibrin formation, thrombin inhibition is a potent way to prevent fibrin generation[26]. We used a high dose (2 mg ml$^{-1}$) of a thrombin inhibitor, argatroban[27], in EDTA or citrated blood and PPP, and found that both blood and PPP droplets (20 µl) quickly rolled off the SHP CNF surface at a very small angle without generating any fibers (Fig. 2b). SEM imaging further confirmed the absence of aligned long fibrin fibers on the surface after sliding by anti-thrombin containing blood or plasma (Supplementary Fig. 7b, c). Similarly, fibrin formation was not detected in the touch-lift test with such anti-thrombin treatment (Supplementary Fig. 7d–f). These findings collaboratively confirmed the identity of the fibers as fibrin.

Fibrin fibers generated at the receding side (Fig. 1d) greatly affected the sliding dynamics of blood/plasma droplets on our SHP CNF surfaces. First, fibrin fibers behaved similar to micro-strings pulling on the blood/plasma droplets, retarding the rolling-off motion and increasing the adhesion of blood/plasma droplets on the surface (Fig. 2c). The roll-off angle (RA) for blood and PPP droplets (mean ± SD) were 28.1 ± 11.6° and 15.0 ± 6.6°, respectively (Fig. 2d). When fibrin generation was inhibited by anti-thrombin, blood and PPP droplets could roll off at a smaller RA of 4.3 ± 1.4° and 2.6 ± 1.3°, respectively. Due to the dose effect of anti-thrombin, some micro-fibrin fibers may still exist at the liquid–solid interface, affecting the receding CA and leading to a large CAH (CAH > RA) for blood/PPP droplets with anti-thrombin (Fig. 2d). Second, the blood or plasma droplet formed a protrusion at its receding region near the surface under the tugging force from fibrin fibers (Fig. 1d and Supplementary Fig. 3), generating a nominal receding angle $\theta_{r\_nom}$ that was smaller than the actual receding angle $\theta_r$ (Fig. 2c). Consequently, the droplet demonstrated a large nominal CA hysteresis $CAH_{nom}$ ($CAH_{nom} = \theta_a - \theta_{r\_nom}$, actual $CAH = \theta_a - \theta_r$, and $\theta_a$: advancing angle). This is not customarily observed on SHP surfaces, which typically have low RAs corresponding to low contact-angle hysteresis[22,28–30]. In a recent report, a similar distorted receding edge for blood droplets sliding down an SHP surface was observed but not explained[22].

Regarding the generation mechanism, fibrin fibers were initiated upon the blood/plasma contact with CNFs. Exposure to CNFs would trigger an extrinsic coagulation cascade reaction, causing the formation of thrombin from prothrombin, and subsequently converting fibrinogen into fibrin monomer[26]. Fibrin monomers would attach onto CNFs and then self-polymerize into long insoluble fibrin fibers, which became visible upon blood/plasma-substrate separation (Fig. 1b, c). An interesting and unexpected observation in this study was that fibrin fibers still generated, despite the presence of anticoagulant EDTA or 3.8% sodium citrate (Fig. 1b and Supplementary Figs. 3 and 4), demonstrating its potency in expediting fibrin formation. In addition, CNFs were reported to activate platelet[31], bind serum albumin and fibrinogen[32], and activate the serum complement system[32,33], which would further trigger coagulation[34]. Electric charges concentrated on sharp geometries, such as on the tips of long CNFs, could also attract fibrinogen adsorption[35] and possibly promote coagulation. However, the detailed mechanism is still unclear and requires further investigation.

CNF's capability to promote fibrin fiber formation was further demonstrated by cultivating EDTA PPP on our CNF surface at 37 °C for 4 min, following published methods[36,37]. A fibrous fibrin meshwork was observed (Fig. 3a); such a fibrous meshwork could be beneficial for hosting activated platelets and blood cells to accelerate coagulation[38–40], making our SHP CNF surface promising for hemostatic application.

**Anti-bacterial property**. Our SHP CNF surfaces demonstrated excellent anti-bacterial properties. We flushed a solution containing *Escherichia coli* (a major infection-causing bacteria[2]) with green fluorescence protein (GFP) expression plasmid over a glass slide that was half-coated with CNFs and nearly no bacteria was found on the SHP CNF surface (Fig. 3b) under the confocal microscope[41] with a 473 nm laser for GFP excitation[42]. The low adhesion of bacteria on our SHP CNF surface is attributed to the low surface energy hydrophobic materials and the micro/nano-roughness[41,43]. This excellent anti-bacteria capability will be beneficial, as it helps keep the hemostatic patch sterile and prevent wound infections[2,32].

**Enhanced clotting without blood loss**. A hemostatic material should promote quick coagulation to minimize blood loss. As a proof-of-concept prototype of using our material as a wound patch, we coated a normal cotton gauze with SHP CNF (Fig. 3c). As cotton could not withstand the high annealing temperature (400 °C) for CNF/PTFE coating, we used CNF/PDMS for coating, taking advantage of the low polymerization temperature of PDMS. As verified previously, the CNF/PDMS surface can promote fibrin fiber generation just like the CNF/PTFE surface (Supplementary Fig. 4d and Supplementary Movies 4 and 5). The cotton gauze, which was initially superhydrophilic and blood absorbing (Supplementary Fig. 9), became SHP after the CNF/PDMS coating (Fig. 3c).

Clotting performance of this SHP CNF gauze was then evaluated. Twenty microliters of the blood, placed between two pieces of gauzes (Supplementary Fig. 10a), were allowed to coagulate for a fixed period of time. Coagulation was terminated by adding 10 ml deionized (DI) water[2,8,15]. Free hemoglobin from red blood cells, not trapped in the clot, would be released into water. A lower hemoglobin level would indicate faster clotting[2,8,15]. The CNF gauze was shown to have a lower hemoglobin level and thus faster clotting compared with normal gauze at 3 min (Fig. 3d).

The non-wetting property of our SHP CNF coating can prevent blood loss at the wound site, by keeping blood within the wound. This feature was demonstrated in vitro, with a silicone tube filled with blood that had a hole opened on its side to mimic a bleeding wound. Cotton gauzes, with and without SHP CNF coating, were used to cover the holes (Supplementary Fig. 10c). The SHP CNF gauze achieved clotting without blood loss, whereas the normal cotton gauze experienced severe blood seepage (Fig. 3e). Therefore, owing to the CNF coating's synergetic capability of promoting fibrin formation and minimal wetting (superhydrophobicity[20,22,44]), our material design strategy can achieve fast clotting without blood loss. This performance can be especially beneficial for chronic bleeding disorders[45].

Furthermore, the air plastron trapped on the SHP CNF surface can be a functional component of the SHP wound patch, as it can help retain the non-wetting feature under high pressure[46]. Without an impervious plastic membrane (Fig. 3e), a single layer of CNF gauze could withstand a pressure of 4.9 ± 0.3 mmHg (mean ± SD) without blood infiltration ($n = 3$; Supplementary Fig. 10d); with a impervious plastic membrane applied at the back of the gauze like the plaster, the trapped air plastron could prevent the CNF gauze immersed in blood from wetting even at 300 mmHg (Supplementary Fig. 10e), which was significantly larger than a lotus leaf's non-wetting pressure (about 100 mmHg)[46].

**Facile clot detachment from the CNF patch**. Another unexpected and rather inherent feature of the SHP CNF patch is that the formed clot can easily detach from the CNF gauze by itself upon clot maturation. This is in sharp contrast to existing

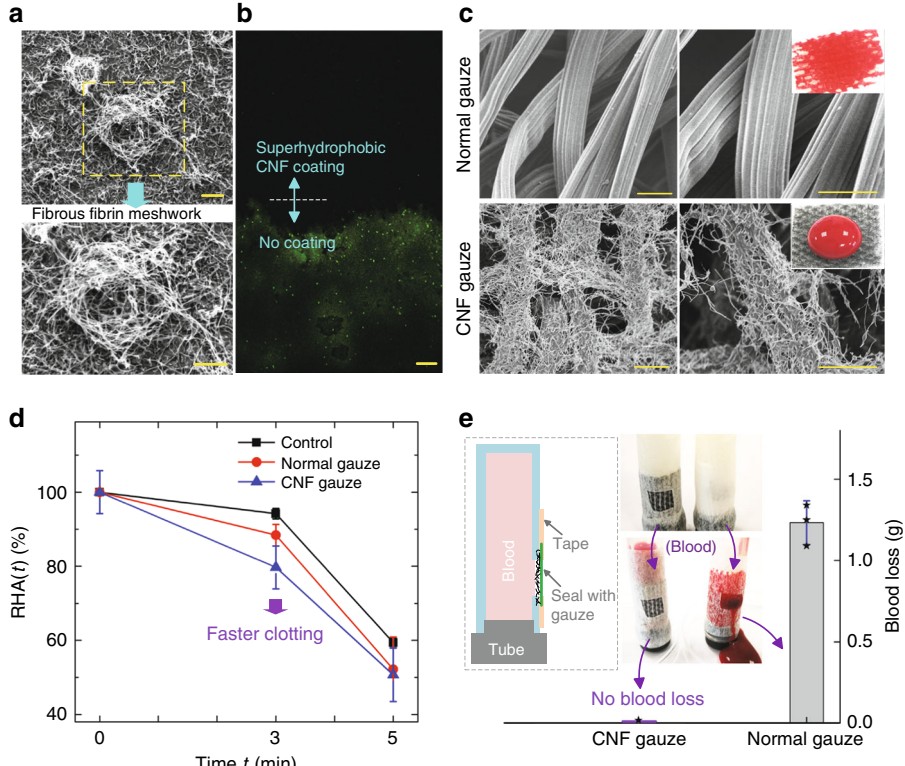

**Fig. 3 Anti-bacterial, rapid coagulation, and non-wetting features. a** Fibrous fibrin meshwork formed on the superhydrophobic CNF/PTFE Ti surface.
**b** Anti-bacterial capability; after flushing with fluorescent bacteria solution, nearly no bacteria (green dots) were observed to attach on the superhydrophobic CNF surface under the microscope with green fluorescent protein excitation. **c** SEM images of the normal gauze and the CNF gauze; the normal gauze is superhydrophilic and blood absorbing; the CNF gauze has a dense layer of CNF coating and repels blood. **d** The relative hemoglobin absorbance RHA($t$) plot, showing the fast clotting performance of the CNF gauze; as the experiment was performed in a petri dish, clotting in the petri dish without any gauze was used as control; the absolute hemoglobin absorbance at clotting time $t$, HA($t$), was measured by the spectrometer at 540 nm ($n=3$); HA(0), the hemoglobin absorbance at $t=0$ min, was used as the reference; the relative hemoglobin absorbance at clotting time $t$, RHA($t$) equals to HA($t$)/HA(0); as the hemoglobin came from unclotted red blood cells, a lower hemoglobin absorbance value would mean faster clotting. **e** Clotting without blood loss ($n=3$); the CNF gauze was used to seal an opening on a silicone tube, mimicking a skin wound covered by a gauze; individual data points are represented by black stars. The CNF coating was in contact with blood in **d** and **e**. Scale bars are 10 μm in **a**, 50 μm in **b**, and 20 μm in **c**. Data in **d** and **e** are shown as mean ± SD, and the error bar represents SD. Source data for **d** and **e** are provided as a Source Data file.

hydrophilic hemostatic dressing materials[1,9,47]; blood will soak and coagulate inside the pores of these materials and the generated clot-dressing composite mixture has strong adhesion to the wound and would be difficult to detach (Supplementary Fig. 9c). Forced peeling of these hydrophilic hemostatic dressings will tear the wound and cause secondary bleeding, complicating subsequent wound care.

We found that the driving force for easy clot detachment from our CNF gauze was the contraction of clot as it matured. In the early stage of coagulation, fibrin fibers (initiated on CNFs, Fig. 4a) would form a fibrin meshwork for clot formation. As such, the clot would have micro-fibrin fibers connected onto the CNFs (Fig. 4c and Supplementary Fig. 12c-3); therefore, a weak connection between the clot and the CNF surface was formed due to the presence of air pockets at the blood–substrate interface (Cassie–Baxter state)[29]. During clot maturation, filopodia from platelets would pull and bend fibrin fibers, generating a contractile stress[48], and causing clot contraction (Supplementary Figs. 11d and 12a). The contractile stress in the clot would pull and remove micro-fibrin fibers adhered on CNFs (Fig. 4b), causing the clot to free itself from the CNF surface after clot solidification and contraction.

Self-peeling behavior of the clot was observed on the rigid Ti mesh substrate coated with CNF after clot contraction (Fig. 4e and Supplementary Fig. 11d-f). On the flexible CNF gauze, clot

peeling was not visibly self-accomplished, as cotton fibers would be deformed by clot contraction; however, the clot could be easily picked off (Fig. 4f). The average clot-peeling tension for our SHP CNF gauze was about $1.7 \pm 1.5$ mN mm$^{-1}$ (mean ± SD), about 54 times smaller than that for the normal hydrophilic gauze ($91.3 \pm 19.4$ mN mm$^{-1}$, Fig. 4f, $n=4$). After clot detachment, CNFs that were initially immobilized on cotton fibers were transferred onto the clot (Fig. 4d and Supplementary Fig. 12c), making the cotton fibers appear smooth (which was drastically different from their original appearance in Fig. 3c) and the detached clot surface appear hairy. This intrinsic clot self-detachment mechanism thus greatly facilitates wound-dressing removal, avoiding wound tear and eliminating secondary bleeding.

We further compared the clot-peeling force of our material with three representative commercial hemostatic products from Smith & Nephew, 3M, and Guardian (Supplementary Fig. 15). These products are marketed as low-adherence or pain-free on removal, but the clot-peeling tension of our material is 24–52 times smaller (Supplementary Fig. 15c; specifically, the clot-peeling tension on our material is about 52 times smaller than Smith & Nephew and about 24 times smaller than 3M or Guardian). Therefore, our material design strategy has brought the clotting peeling force of hemostatic materials to a very low level.

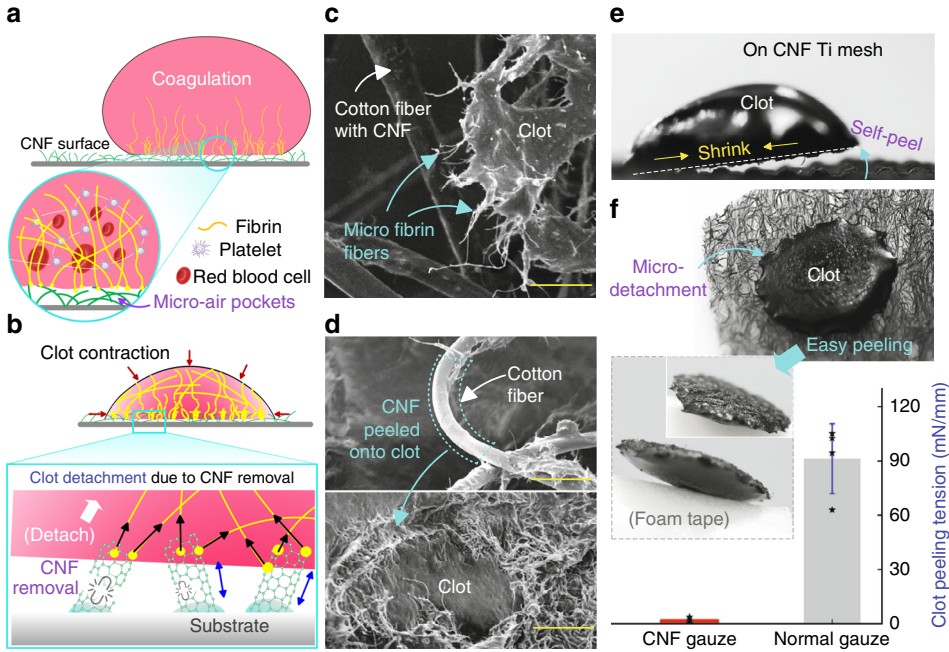

**Fig. 4 Facile clot detachment. a** Illustration of fibrin fibers formation from blood on a superhydrophobic CNF surface. Due to its non-wetting property and the presence of micro-air pockets (Cassie–Baxter state[28,49]), the CNF surface would only partially attach onto the clot. **b** During clot maturation, the fibrin mesh would squeeze out the serum and shrink the clot into a smaller size;[39] contractile stress during clot contraction and solidification would remove CNFs connected with micro-fibrin fibers, causing the clot to free itself from the superhydrophobic CNF surface. **c** SEM image of the micro-fibrin fibers adhered on the CNFs-coated cotton fiber after clot shrinkage (Supplementary Fig. 12a, b). **d** SEM images show CNFs transferred onto clot after clot detachment, resulting in a smooth cotton fiber, and a hairy clot surface (Supplementary Fig. 12c). **e** Clot self-peel from a stiff CNF Ti mesh surface (Supplementary Fig. 11d–f). **f** Facile clot detachment; clot formed on the CNF gauze can be easily picked off by a foam tape and the corresponding clot-peeling tension is about 54 times smaller than that for peeling a clot from the normal gauze ($n = 4$, the setup in Supplementary Fig. 13); briefly, the gauze with a clot was mounted onto a force sensor, which was used to record the clot-peeling force for the calculation of peeling tension; data in **f** are shown as mean ± SD, the error bar represents SD, and individual data points are represented by black stars. Scale bars are 25 μm in **c** and 50 μm in **d**. Source data for **f** are provided as a Source Data file.

**In vivo verification**. To verify the aforementioned features of this new hemostatic material, in vivo experiments were performed on rats with the back-bleeding model[9] (incisions made on the back of rats for gauze application; Supplementary Fig. 16a-d). The normal gauze was blood absorbing, leaving behind an open wound (bottom in Fig. 5a, b). In contrast, no blood was observed to seep through the CNF gauze (Fig. 5a-top), demonstrating its excellent blood-repelling property. Further, a darkened gel-like clot was observed under the CNF gauze at 3 min (Fig. 5b-top), which properly sealed the wound as opposed to the wound that remained open under the control gauze, revealing the CNF gauze's capability in promoting coagulation in vivo. The average blood loss (characterized by the weight increase in gauze at 3 min; Supplementary Fig. 16e, mean ± SD) for the CNF gauze was 0.3 ± 0.7 mg (Fig. 5d), about 1.5% of that for the normal gauze (19.8 ± 9.0 mg), confirming CNF gauze's capability in minimizing blood loss. Thus, in vivo work corroborated our in vitro findings: the blood-repelling CNF gauze could promote coagulation, minimize blood loss, and help achieve a good clot-sealed wound.

The force required for gauze removal was also measured in vivo. As the wound under the normal gauze was torn seriously during peeling (Fig. 5c-right), it was difficult to accurately measure the gauze-wound contact width. The maximum peeling force was used to qualitatively evaluate the performance of our CNF gauze (Supplementary Fig. 16i). The average maximum peeling force (mean ± SD) for the CNF gauze was 7.2 ± 8.6 mN, about 43 times smaller than that for the normal gauze (315.2 ± 61.3 mN, Fig. 5e). Our easy-to-peel CNF gauze did not tear the wound due to CNF detachment

(Fig. 5c top-left, Supplementary Movie 6). Many CNFs that were initially coated on cotton fibers (Fig. 5g) were removed during clot peeling (Fig. 5f), just like in vitro tests (Supplementary Fig. 12c). Moreover, gentle stretching did not cause wound tearing (Fig. 5c left-bottom) or secondary bleeding (Supplementary Fig. 16f). In the control group, the clot strongly bound the wound and even skin hairs (Fig. 5c top-right) to the gauze, forming a stiff clot-gauze-hair concrete (Fig. 5h). Forced peeling would tear the wound and cause secondary bleeding (Fig. 5c top-right, Supplementary Fig. 16f-right, and Supplementary Movie 7), increasing the chance of infection. In vivo findings therefore confirm the remarkable features of our wound-dressing materials (Fig. 5i), which can successfully address the serious problems (blood loss and strong clot adhesion) plaguing the application of conventional hydrophilic hemostatic materials.

Previous studies showed that multi-wall CNFs and the CNF/PDMS composite were non-toxic[50,51], with the 1-week cell viability exceeding 95%[51]. Our material is designed for hemostatic purpose and will be in contact with skin for a short time. We thus conducted in vivo skin compatibility tests by attaching our material (10 mm by 10 mm) onto the rat skin (hair shaved) with the clinically approved tape for 12 h (Supplementary Fig. 17a, b). Compared with the skin under the pristine gauze (control), the skin under our CNF gauze appeared normal and no itching or erythema was observed after 12 h (Supplementary Fig. 17c). Thus, based on previous reports on cell non-toxicity[50,51] and our skin compatibility test, our CNF/PDMS material would be safe for hemostatic use.

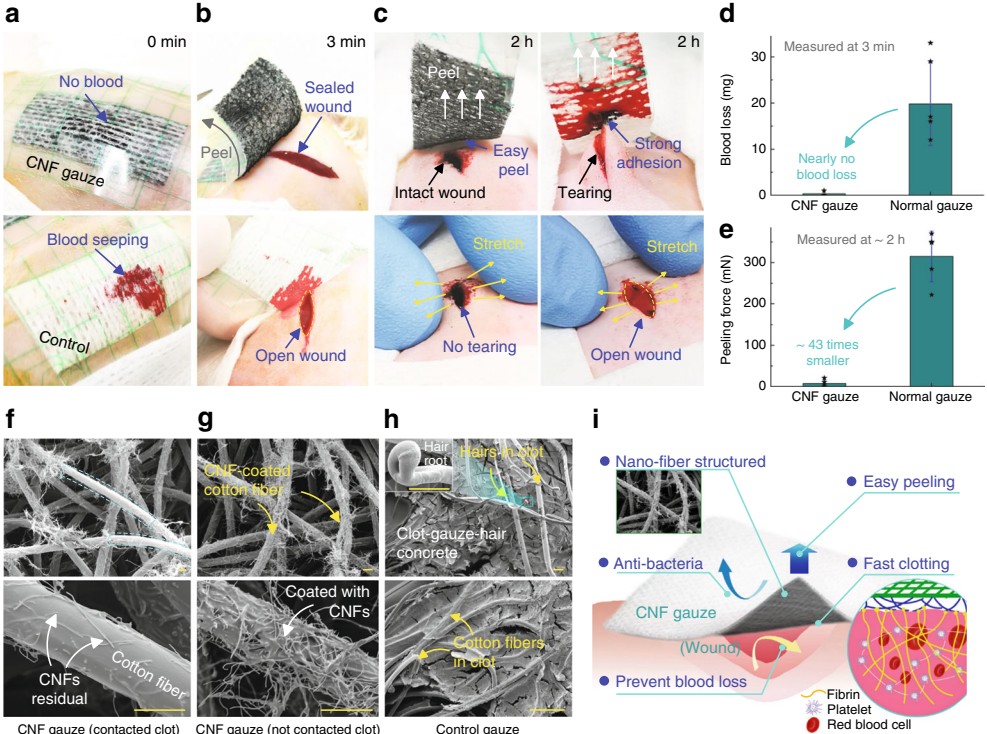

**Fig. 5 In vivo animal experiment. a** The plaster-like gauzes were patched onto incisions on rat back (Supplementary Fig. 16a-d); the control cotton gauze got wet quickly, while the CNF gauze prevented blood loss. **b** Peeling the gauze at 3 min to measure the blood loss; the CNF gauze helped form a gel-like clot, which properly sealed the wound; under the control gauze, an open wound was observed. **c** Peeling the gauze at about 2 h to measure the peeling force (Supplementary Fig. 16g-i); the CNF gauze could be easily peeled off and gently stretching the wound did not cause wound tearing or bleeding (Supplementary Movie 6); in contrast, peeling the normal gauze caused wound tearing and bleeding (Supplementary Movie 7). **d** The CNF gauze minimized blood loss ($n = 6$). **e** Peeling force for the CNF gauze was significantly smaller than that for the normal gauze ($n = 5$). **f** SEM images of the area in contact with blood on the CNF gauze in **c**; CNF residuals after clot detachment were observed on cotton fibers. **g** SEM images of the area not in contact with blood on the CNF gauze in **c**, where cotton fibers were densely coated with CNFs. **h** SEM images of the peeled normal gauze in **c**, showing a clot-gauze-hair concrete, with rat skin hairs imbedded in clot; a hair root is shown in the inset, implying that skin hairs that were stuck in the clot were pulled out from skin during gauze peeling. **i** Schematic of the hemostatic CNF gauze/plaster for wound treatment. Data in **d** and **e** are shown as mean ± SD, the error bar represents SD, and individual data points in **d** and **e** are represented by black stars. Scale bars are 10 μm in **f**, 15 μm in **g**, and 100 μm in **h**. Source data for **d** and **e** are provided as a Source Data file.

## Discussion

We have developed and demonstrated a strategy for the design of wound-dressing materials featuring both rapid coagulation and facile clot removal, based on superhydrophobicity imparted by surface-immobilized CNFs. The developed SHP hemostatic CNF gauzes are shown to simultaneously combine a host of therapeutic functionalities. First, they achieve fast clotting by expediting the formation of micro-fibrin fibers. Second, clotting without blood loss can be achieved, owing to the pressure-resistant non-wetting feature of an SHP surface[28,46,52]. Third, there is a minimal contact between clot and our SHP CNF surface; driven by the contractile stress in the clot contraction phase[48], the SHP CNF hemostatic surface will inherently tend to detach from the clot upon clot maturation, thus allowing a natural and unforced removal of the hemostatic dressing without causing secondary bleeding. The proposed hemostatic and clot-release mechanism with a SHP CNF surface alleviates the serious problems of blood loss and strong clot adhesion plaguing the application of hydrophilic hemostatic materials[8,9,24]. Forth, the SHP CNF patch can significantly reduce bacterial attachment, reducing infection risks[24]. Finally, the CNFs are immobilized in a polymeric matrix on the gauze substrate, preventing free micro/nanoparticles or fibers from entering the vascular stream; also, skin compatibility tests show that our material can be safe for hemostatic use.

Different from existing hemostatic products/materials, our material works in a distinctive manner: stop bleeding first, reinforce clotting subsequently, and enable unforced natural detachment in the end. This offers the following advantages. First, the SHP feature helps stop bleeding immediately upon application. Bleeding therefore ceases before the formation of a strong clot. This can be life-saving for hemorrhage such as in severe accidents and military combats[53], as it takes time for clotting to occur even with the most effective hemostatic materials such as QuickClot™ and Hemogrip™ [7,54]. Second, rapid clotting is achieved by the nano-engineered surface structure, without the help of active clotting agents/chemicals. Our nano-engineered fibrous structure promotes quick growth of fibrin network to seal the wound. Clotting agents are routinely used in hemostatic products, such as the hydrophobically modified chotisan[54–56], which may provide shorter clotting time than our material. However, our material brings it with a distinct strategy of hastening coagulation using a nano-structuring approach also ensuring facile removal, a critical feature to avoid secondary bleeding (a quantitative comparison between our material and the hydrophobically modified chitosan[54–56] is provided in Supplementary Fig. 18). Our material yields a reduction of the clot peeing force, by a good order of magnitude. Compared with commercial hemostatic products, the clot-peeling tension of our material is about 24–52 times smaller (Supplementary Fig. 15c). This natural and facile clot detachment

feature is unique to our SHP hemostatic material and has not been reported earlier according to our knowledge.

Our work therefore pioneers a strategy for designing more efficient hemostatic material using surface nano-engineering. We demonstrate that a nano-structured surface can achieve fast clotting with unique blood loss-free, unforced detachment from the wound site, and also reduced bacteria adhesion. The multifunctional material concept presented in this work shows clear potential to significantly advance the state-of-the-art of wound dressings, bringing benefits to common wounds, surgery, and even hemophilia. In the current study, we used non-biodegradable carbon nanofibers to design the nano-engineered surface. Biodegradable nanofibers could be developed for internal use. Also, clot-promoting performance can be further enhanced by performing a parametric study of the surface topographical features and chemical composition. With these endeavors, the material design strategy developed here will lead to a more efficient hemostatic product for clinical use in the near future.

## Methods

**Superhydrophobic carbon nanofiber coating**. CNFs[18,57] (purity: 98%, diameter: 100 nm, length: 20–200 μm, Sigma Aldrich) were pre-mixed with PTFE powder (1 μm particle size, Sigma Aldrich)[19] or PDMS (Slygard 184)[21] and were applied on substrates via spray coating[58]. Before spray coating on a flat Ti substrate (Ti6Al4V, 1 mm thick), the Ti mesh (#60; nominal aperture: 250 μm; wire diameter: 125 μm; Supplementary Fig. 11a), or transparent glass slides (25 mm by 75 mm, 1.1 mm thick), these substrates were ultrasonically cleaned (10 min) with acetone and isopropanol, and were further cleaned with oxygen plasma. The cotton woven gauze (Smith & Nephew Pte Ltd; Supplementary Fig. 9a, b) was used as received. Before spray coating, CNF/PTFE or CNF/PDMS composite dispersion in dichloromethane was first prepared by mixing the CNF and dichloromethane dispersion with the PTFE and dichloromethane or PDMS and dichloromethane dispersion. The CNF and dichloromethane dispersion was prepared by dispersing CNFs in dichloromethane with a probe ultrasonicator for 1 min; the PTFE and dichloromethane or PDMS and dichloromethane dispersion was prepared by dispersing the PTFE powder or PDMS (pre-polymer to cross linker weight ratio 9:1) in dichloromethane under ultrasonication for 20 min. The CNF and dichloromethane dispersion was then mixed with the PTFE and dichloromethane or PDMS and dichloromethane dispersion, and sonicated for 10 min to prepare the CNF/PTFE or CNF/PDMS composite dispersion in dichloromethane. The composite dispersion was spray-coated onto sample substrates (pressure: 430 kPa). The spray-coated CNF/PTFE sample was baked for 30 min at 400 °C in a low oxygen environment to prevent oxidization; the CNF/PDMS sample was baked at 80 °C for 1 h. As the cotton could not withstand the high temperature (400 °C) for PTFE coating, the cotton gauze was coated with CNF/PDMS instead. The CNF/PTFE nanocomposite had a CNF to PTFE weight ratio of 1:9; the CNF/PDMS nanocomposite had a CNF to PDMS weight ratio of 1:2, whereas a higher concentration of PDMS would cause CNF agglomeration and result in significant coverage of the exposed CNF surface by PDMS, leading to RAs larger than 10°. The effect of CNF concentration on superhydrophobicity was also investigated by spray coating low-concentration CNF on Ti substrates (HP #1: only PTFE; HP #2: 0.2 wt% CNF in PTFE) following the same protocol (Supplementary Fig. 1a, b). To remove any loosely attached CNFs, we exposed samples to compressed $N_2$ gas from a spray gun (nozzle diameter: 0.8 mm; pressure: 430 kPa; sample to nozzle distance: ~15 cm) before test to make sure there was no free CNF on our material.

**Blood and platelet poor plasma**. Fibrin fiber generation tests (Figs. 1 and 2, and Supplementary Figs. 3–7) were performed using blood and PPP with the presence of EDTA or sodium citrate. Fresh EDTA porcine blood, ordered from SingHealth Experimental Medical Centre (Singapore), was collected into sterile vacutainer tubes with K2 EDTA (3.0 ml, purple) and transported in a sealed foam container with ice. PPP was obtained by centrifuging the blood in EDTA tubes first at 1800 G for 10 min (Sorvall ST 8 Centrifuge; room temperature 20 °C) and then at 3000 G for another 10 min[59]. As EDTA would prevent blood coagulation by irreversibly chelating calcium ions, EDTA blood was not suitable for coagulation test; for in vitro clotting tests (Figs. 3 and 4), 3.8% sodium citrate was used as the anticoagulant[2,8,24]. Citrated porcine blood from the same vendor was collected into polypropylene tubes by mixing blood with 3.8% sodium citrate solution at a volume ratio of 9:1[37] and was transported following the same protocol.

**Contact angle, roll-off angle, and surface morphology**. CAs and RAs were measured using a custom-built device. CA was measured using the sessile method by dispensing 5 μl liquids (DI water, blood and PPP) on flat substrates (the CNF/PTFE Ti and CNF/PDMS Ti surfaces) and 20 μl liquids on the CNF Ti mesh. Dynamic CAs, including advancing CA $\theta_a$, receding CA $\theta_r$, CA hysteresis CAH

($CAH = \theta_a - \theta_r$), nominal receding CA $\theta_{r\_nom}$, nominal CA hysteresis $CAH_{nom}$ ($CAH_{nom} = \theta_a - \theta_{r\_nom}$), and RA $\theta$ were measured using the tilting method[60], by placing a 20 μl droplet (for water, blood, and PPP with or without anti-thrombin) on the sample surface and tilting the sample till droplet roll-off. For PPP or blood on the SHP CNF surface, due to the existence of fibrin fibers, RA $\theta$ was defined as the tilt angle when fibrin fibers fractured and the droplet rolled off quickly. Static and dynamics CAs were averaged over four repetitions ($n = 4$). Surface morphology of the prepared samples and the surfaces after blood/plasma tests were characterized by SEM (SEC, SNE-4500M) after gold coating.

**Nominal receding angle**. The nominal receding angle $\theta_{r\_nom}$ for blood or PPP droplets (20 μl) on the SHP CNF surface was measured at the moment before catastrophic fibrin fiber fracture and droplet rolling down (Supplementary Fig. 8a). $\theta_{r\_nom}$ was the angle between the liquid–substrate interface AB and the straight portion of the liquid–air interface CD that was above the droplet protrusion (Supplementary Fig. 8b). The acquired high-resolution image (1920 × 1080 pixels) was smoothed via pixel intensity averaging and reducing pixel resolution to 480 × 270 pixels, making it easier to localize the liquid–air or the liquid–substrate interface (lines AB and CD; Supplementary Fig. 8c, d). A reference circle (radius: 30 pixels) was superimposed onto the image to intersect the straight portion of the liquid–air interface at two points C and D. The position of D was further confirmed by the pixel greyscale value; point D has an abrupt change in the gray value (Supplementary Fig. 8e, f). The position of C was confirmed similarly. The line AB on the liquid–solid interface was also determined using the same method. The aforementioned procedures were performed manually and would have sufficiently high accuracy, with errors in the order of the size of one pixel, which would be equivalent to an angle error of about 0.95° (arctan(1/60)).

**Anti-thrombin test**. The anti-thrombin, argatroban[27] (Purity: > 98%, BioChemPartner lt, China) was first dissolved in 0.9% NaCl solution and then added into EDTA blood or PPP to achieve a final dose of 2 mg ml$^{-1}$, which was higher than the value used in animal studies to ensure sufficient thrombin inhibition[61].

**Porcine fibrin ELISA test**. The porcine fibrin ELISA kit was ordered from MyBioSource, Inc. (Catalog #MBS261977)[25]. Sliding tests with EDTA PPP droplets were respectively performed on three SHP CNF/PTFE and CNF/PDMS Ti surfaces (15 mm by 7 mm); fibrin fiber generation during the sliding test was confirmed using the setup in Supplementary Fig. 2. After sliding test, the standard diluent from the kit was flushed over the surfaces using a pipette into reaction wells. As a control, the same procedures were performed on three pristine SHP CNF/PTFE Ti surfaces. Then, ELISA test was performed following standard instructions. Optical density reading (450 nm) was used to calculate the fibrin concentration washed into the reaction wells from the tested SHP surface, with reference to the standard curve (fibrin concentration to optical density) acquired with porcine fibrin standard samples[25], to verify the existence of fibrin on the CNF surfaces after PPP sliding tests.

**Static fibrin growth**. Forty microliters of EDTA PPP was dispensed onto a SHP CNF/PTFE Ti surface placed in a plastic petri dish and cultivated for 4 min at 37 °C; the reaction was terminated by slowly adding sufficient DI water into the petri dish;[36] the samples were then carefully rinsed by dipping into DI water three times[36,37]. Samples were air-dried; fibrin structures grown on the surface were observed under SEM after gold coating.

**Anti-bacteria test**. For convenient observation of bacteria attached on the surface, a SHP CNF/PTFE coating was half-coated on a glass slide following the same procedure for the CNF/PTFE Ti surface. The SHP CNF/PTFE glass surface and the CNF/PTFE Ti were shown to have the same performance through water, blood, and PPP testing. Ten microliters of glycerol stock (50% glycerol, 50% cell culture in Luria–Bertani (LB)) of E. coli MG1655 (K-12) (ATCC®700926™), harboring constitutive GFP expression plasmid stored at −80 °C, was added into 3 ml fresh LB broth, supplemented with Kanamycin (Km) (50 μg ml$^{-1}$). The cells were incubated in a shaking incubator at 37 °C with a shaking speed of 225 r.p.m. The culture was diluted to OD$_{600}$ of 0.5 with fresh LB (50 μg ml$^{-1}$ Km). The CNF/PTFE glass slide sample was first sterilized with ultraviolet, then 40 μl cell culture was flushed over the sample surface across areas with and without CNF coating. The sample was subsequently air-dried in Biosafety Cabinet (Gelman, Singapore) for 20 min. Bacteria attached on the CNF surface were observed under the confocal laser scanning microscope (Olympus confocal FV1200, Japan), with laser wavelength of 473 nm for exciting the GFP[42].

**Chitosan gauze**. To compare the clotting performance of our CNF gauze, chitosan gauze was prepared by spray coating the chitosan and PDMS composite dispersion onto the pristine gauze. For fair comparison, weight ratio of chitosan to PDMS was 1:2, the same as CNF to PDMS weight ratio for our CNF gauze. Twenty-five milligrams of chitosan (medium molecular weight, Sigma Aldrich) was first dissolved in 5 ml 0.15 M acetic acid with an ultrasonic probe; 50 mg PDMS (pre-polymer to cross linker weight ratio 9:1) was dispersed in 10 ml acetone by the

ultrasonic probe. After mixing the two dispersions under ultrasonication for 5 min, the composite dispersion was spray-coated onto a pristine gauze (size: 10 cm by 10 cm) following the same protocol for preparing the CNF gauze. After coating, the chitosan gauze was baked at 80 °C for 1 h to evaporate the solvent.

**In vitro clotting test**. With reference to published studies[2,8,24], clotting test was performed using the citrated blood. Different types of gauzes (the SHP CNF gauze and the normal superhydrophilic gauze; size: 15 mm by 15 mm) were pre-warmed in 20 ml polystyrene (PS) plastic petri dishes (37 °C water bath). After mixing the citrated blood with 0.2 M CaCl$_2$ at a volume ratio of 10:1 to initiate coagulation[24], 20 μl blood was immediately dispensed and sandwiched between two gauzes in the petri dish (Supplementary Fig. 10a; on the CNF gauze, blood was in contact with the surface coated with CNF). Blood dispensed in an empty petri dish without gauze was used as the control case (denoted as Control in Fig. 3d). Blood dispensed on different samples were allowed to coagulate at 37 °C for 0, 3, and 5 min, terminating coagulation by adding 10 ml DI water into the petri dish without disturbing the clot. Non-clotted red blood cells (free blood cells not trapped in the clot) would hemolyze and release hemoglobin into the water[15,24]. Optical absorbance of the resulting hemoglobin solution at different clotting time $t$, HA($t$), measured by the spectrophotometer at 540 nm, would represent the amount of hemoglobin from unclotted red blood cells; HA(0), the absolute hemoglobin absorbance of 20 μl blood in 10 ml DI water at $t = 0$ min, was used as reference; RHA($t$), the relative hemoglobin absorbance at clotting time $t$, was calculated as HA($t$)/HA(0) (averaged over three repetitions, $n = 3$; using the same batch of blood)[8]. As the hemoglobin came from unclotted red blood cells, a smaller RHA($t$) would mean faster clotting[2,8,24].

As for the nominal blood contact area (NCA) during the clotting test, following the same protocol, 20 μl blood was placed between two normal gauzes or two CNF gauzes for coagulation to occur. After 1 h, the two pieces of gauze were separated and a picture taken from the top was processed to calculate the NCA (averaged over three repetitions; $n = 3$); NCA on the normal gauze was considered as the red-color area, which was soaked by blood, whereas NCA on the CNF gauze was considered as the clot-gauze contact area (Supplementary Fig. 10b). In this test, the uncoated gauze absorbed quickly, leading to a large blood contact area, whereas the SHP CNF gauze repelled blood and its blood contact area was only 14.2 ± 0.7% (mean ± SD; $n = 3$) of that on the uncoated pristine gauze (Supplementary Fig. 10b). Following the same protocol, clotting performance of our CNF gauze was further compared with the chitosan gauze with 50 μl blood ($n = 3$); the results (Supplementary Fig. 14) show that our CNF gauze can have a better clotting performance than the chitosan gauze.

**Clotting without blood loss**. Mimicking a wound dressed with a medical gauze, an opening representing the wound (8 mm by 5 mm) was made on a silicone tube (inner diameter: 7.8 mm) and was covered with the SHP CNF gauze by medical tape (Supplementary Fig. 10c). A silicone tube dressed with the normal gauze was used as control. The citrated blood was mixed with CaCl$_2$ to initiate coagulation. Blood (1.5 ml) was quickly filled into the silicone tube and placed in a petri dish. After 10 min at room temperature, the tube was removed from the petri dish and the weight increase in petri dish due to blood leakage was measured as the blood loss. The test was repeated three times ($n = 3$) to get the average blood loss.

**Blood non-wetting tests of the CNF gauze**. To test the maximum pressure that the CNF gauze (without a back-supporting impervious membrane) could withstand before blood leakage, citrated blood was slowly filled into a long tube with a hole sealed by the CNF gauze (Supplementary Fig. 10d). The height of blood column $h$, which was measured at the moment of blood droplets leaking through the CNF gauze, was used to calculate the maximum anti-leakage pressure $P$ for a single layer of CNF gauze without the back-supporting impervious membrane ($P = \rho g h$, where $\rho$: blood density, $g$: gravitational constant; averaged over three repetitions; $n = 3$).

Anti-wetting property of the SHP CNF gauze with a back-supporting impervious membrane was further tested using the setup in Supplementary Fig. 10e. A transparent PS petri dish was glued (epoxy glue, Bostik) onto the back of the SHP CNF gauze as the impervious membrane; glue was not applied at the central part of the gauze. Subsequently, the petri dish with the attached CNF gauze was placed in a transparent pressure chamber and filled with blood to completely immerse the CNF gauze (Supplementary Fig. 10e); citrated blood was used to ensure blood fluidity. Hydrostatic pressure applied on the CNF gauze surface was controlled by pumping air into the sealed pressure chamber with the assistance of a check valve and a pressure sensor. As the CNF gauze was placed 2–3 mm deep in the blood, the air pressure in the pressure chamber was taken as the hydrostatic pressure exerted onto the CNF gauze. A camera projecting upward was used to detect whether the central part of the CNF gauze was wetted by blood at a given pressure.

**Clot self-detachment**. The SHP CNF Ti mesh and the SHP CNF gauze placed in a 20 ml PS plastic petri dish were pre-warmed at 37 °C (Supplementary Fig. 10a). After mixing citrated blood with 0.2 M CaCl$_2$ at a volume ratio of 10:1 to initiate coagulation, 100 μl blood was immediately dispensed onto the CNF Ti mesh or the

CNF gauze for the clot to form. After natural solidification and drying overnight, clots on the SHP CNF surfaces, before and after detachment, were coated with gold for SEM observation.

**Clot-peeling force**. Citrated blood was mixed with 0.2 M CaCl$_2$ at a volume ratio of 10:1 to trigger the coagulation. Blood clot was allowed to form and solidify overnight at 37 °C on the CNF gauze surface or between two pieces of uncoated gauze by dispensing 20 μl blood on the CNF gauze surface or onto two stacked normal gauzes (Supplementary Fig. 13). Using the setup in Supplementary Fig. 13a, b, the force required for peeling the clot from the CNF-coated gauze or the uncoated gauze was measured[62,63]. Briefly, the backside of the gauze with solidified clot was first mounted onto the force sensor (capacity: 980 mN; resolution: 0.1 mN) by 3 M high-strength adhesive tape; the clot on the CNF gauze was peeled from one side by pulling a thin cotton wire which was glued onto the clot by epoxy (Supplementary Fig. 13a); the two uncoated gauzes adhered together by clot were peeled from each other by pulling the top gauze (Supplementary Fig. 13b). Pulling motion was performed with a homemade translation stage at 0.5 mm/s and the force data were recorded by a National Instruments multifunction data-acquisition device (NI USB-6218) controlled by Labview. As the peak peeling force $F_{max}$ (Supplementary Fig. 13c) would occur at the maximum clot width $W$ (Supplementary Fig. 13d), the normalized clot-peeling tension $F_{max}/W$ (averaged over three repetitions) was used to compare the clot-peeling tension for the CNF gauze and the uncoated normal gauze.

Clot-peeling force of our material was further compared with three representative commercial hemostatic products (Supplementary Fig. 15a). Product #1 was the dressing from Smith & Nephew, labeled to have "minimal pain on removal", #2 was a pad from 3 M, labeled to have "pain-free removal", and #3 was a product from Guardian, labeled to be a "non-adherent pad for easy removal". The Nexcare hydrophilic dressing material from 3 M was used as the substrate material for clotting to occur (Supplementary Fig. 15b). Experimental procedures are described in Supplementary Fig. 15b. The Nexcare substrate material (25 mm wide and 100 mm long) was adhered onto the stainless base; citrated blood mixed with 0.2 M CaCl$_2$ solution at a volume ratio of 10:1 was then dispensed onto the substrate to soak it with blood. Hemostatic samples (30 mm long and 10 mm wide) were immediately placed onto the substrate material, allowing the clot to form between the hemostatic sample and the substrate material. Clotting was first allowed to occur at 37 °C for 1 h; clot solidification was subsequently accelerated by exposing the clotted samples to warm air flow from a hair dryer for 30 min. After that, samples were peeled from one side to measure the peeling force (Supplementary Fig. 15b). Peeling tension was calculated as the maximum peeling force divided by the sample width (repeated for three times; $n = 3$). Besides, WCA was measured with 10 μl DI water ($n = 5$) to provide information on hydrophobicity for different samples (Supplementary Fig. 15c).

**Preparation of the plaster-like CNF gauze for in vivo test**. CNF gauze and the normal uncoated gauze (used as control) were prepared to be plaster-like for in vivo experiment (Supplementary Fig. 16a, b). The gauze was cut to be 35 mm long and 15 mm wide, and was mounted onto a transparent adhesive tape (50 mm long and 25 mm wide; Guardian Adhesive Transparent Film Roll). An opening (10 mm long and 5 mm wide) was made in the center of adhesive tape (Supplementary Fig. 16a), allowing blood to seep through the gauze in case of excessive bleeding. Prepared CNF gauze and normal gauze were sterilized under UV for 30 min before experiment.

**In vivo experiment**. Female rats (Sprague–Dawley, 11–13 weeks old, average body weight: 255.6 ± 19.7 g, mean ± SD) were ordered from InVivos Pte Ltd, Singapore, with approval from the National University of Singapore Institutional Animal Care and Use Committee (NUS IACUC protocol number: R18-0961), complying with all relevant ethical regulations for animal testing and research. Rats were anesthetized by isoflurane (4–5% in 100% oxygen for induction and 1–3% for maintenance) with thermal pad to prevent hypothermia. Hair on the back was first shaved with an electrical razor; residual hair was removed with Veet® hair removal cream[64]. Surgical instruments were autoclaved; experiment was performed in the biological safety cabinet using aseptic techniques. The operative site on rat back was disinfected with iodine and 70% v/v ethanol successively for three times. Two incisions (about 1 cm long; cut down to muscle) were made on the back of the rat, one on the left and one on the right at the same position; upon incision, the plaster-like gauze was applied onto the wound instantly (Supplementary Fig. 16c, d).

As for the blood loss, it was characterized by measuring the weight increase in gauze. Weight of the gauze peeled at 3 min minus its initial weight was taken as the blood loss. Gauze weight was measured by a high-precision weighing balance (resolution: 1 mg). The data were averaged over six repetitions.

As for the peeling force, it was measured by peeling the gauze along the wound after about 2 h (132 ± 10 min and 133 ± 10 min for the control and the CNF gauze, respectively; mean ± SD, $n = 5$), allowing the clot to sufficiently mature and solidify. Before gauze peeling, the adhesive film around the gauze was carefully trimmed away (Supplementary Fig. 16g), otherwise the adhesive film taped on skin would interfere with the peeling force. After adhesive film trimming, one end of the gauze was adhered onto a metal hook by high-strength tape (Supplementary

Fig. 16h), which was attached onto a force sensor (capacity: 980 mN; resolution: 0.1 mN) by a thin wire. Peeling was assisted by a vertical numerical translation stage and the peeling force was recorded. The maximum force during peeling (averaged over five measurements; Supplementary Fig. 16i) was used for a quantitative comparison.

Skin compatibility test was also conducted in vivo by attaching our material onto rat skin for 12 h (Supplementary Fig. 17). Preparation of the samples used for skin compatibility test is illustrated in Supplementary Fig. 17a, with a 10 mm by 10 mm CNF sample attached onto a 20 mm by 20 mm pristine gauze; this design allowed us to compare two adjacent skin areas contacted with different gauze materials. Before test, rats were anesthetized and shaved; prepared samples were applied on rat skin with transparent clinical tape for 12 h (Supplementary Fig. 17b; tested on three rats); to prevent samples from being scratched off by rats, rat jackets (InVivos Pte Ltd) were used to protect the gauze samples. After 12 h, rats were anesthetized to remove the gauze; the tested skin area was gently cleaned with 70% v/v ethanol to examine any difference between the areas under our CNF gauze and the pristine gauze (Supplementary Fig. 17c).

**Reporting summary**. Further information on research design is available in the Nature Research Reporting Summary linked to this article.

## Data availability

All experimental data within the article and its Supplementary Information are available from the corresponding authors upon reasonable request. The source data underlying Figs. 2a, d, 3d, e, 4f, 5d, e, and Supplementary Figs. 1a, 10b, 14 and 15c are provided as a Source Data file.

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

## Acknowledgements

This work was supported by Singapore National Research Foundation (NRF; grant number NRF2017-ITS002-012), the Singapore National Medical Research Council (NMRC; grant number NMRC/OFIRG/0060/2017), the Singapore Ministry of Education AcRF Tier 1 Grant 2018 (PI: Yap), and the European Union's Horizon 2020 research and innovation program (grant number No 801229). We acknowledge Jingyun Zhang for his help on the anti-bacteria test and Kum Cheong Tang for the assistance on SEM measurement.

## Author contributions

Z.L., A.M., D.P. and C.H.Y. designed the experiments. A.M. and L.C. prepared samples. Z.L. performed wetting, SEM, fibrin fiber verification, and blood/plasma tests. M.Y. and F.T. performed fibrin fiber generation test and the porcine fibrin ELISA test. Z.L. and Z.Y. performed animal experiment. Z.L., A.M., Z.Y., D.P. and C.H.Y. analyzed the data. Z.L., A.M., Z.Y., D.P. and C.H.Y wrote and proof-read the paper.

## Competing interests

The authors declare no competing interests.
