## [Peer Review File · Nature Communications]

Reviewers' Comments:

Reviewer #1:

Remarks to the Author:

The authors report the design of wound dressing materials based on super-hydrophobicity imparted by surface-immobilized CNFs. They reported that CNFs promote quick fibrin growth and cause rapid clotting. This will attract interests from readers, however, there are some points that should be clarified.

1. The authors mentioned that " active clotting agent pose a safety threat of micro-thrombosis, however, how about the materials used to form PDMS? PDMS requires crosslinker to proceed polymerization, which is also have threat to human body.
2. Similar research is reported by Jun-Yon Park et al., "Biodegradable polycaprolactone nanofibers with beta-chitosan and calcium carbonate produce a hemostatic effect"(Polymer 123(2017)194-, however, the authors do not refer the paper. The authors should compare their results with previous report at least, and should mention their intensive points or advantages.
3. Please clarify how the authors obtained "Hemoglobin absorbance" in Fig.3.

Reviewer #2:

Remarks to the Author:

This manuscript by Li et al reports a hemostatic material with multiple advanced properties such as superhydrophobicity, promoting blood coagulation, antibiosis and facile detachment from wound. It's both interesting and practical for broad audience. I am not an expert on medical science and found it difficult to catch the state-of-art advances on wound treatment and clinical gauze. But I'm concerned that for a gauze material, important factors are biocompatibility and biotoxicity, which are missing in the manuscript.

1. I think no doctor will attach excessive bleeding wound with gauze only, antibacterial medicines will be applied on the wound. So superhydrophobic feature is nice for gauze materials, it's helpful on holding bleeding, but it is not major improvement.
2. The biocompatibility and biotoxicity of the superhydrophobic gauze material need to be investigated. To be noted PTFE is non-degradable, is there antigenic reaction if PTFE particles come inside the blood vessel? And what about PDMS and CNF?
3. The authors spent substantial words to describe the fibrin formation when blood contacts their material, but Hemoglobin absorbance-time plot doesn't show a substantial improvement on the effect of coagulation. Please explain why?
4. The authors claim the sample has a low actual contact angle hysteresis (CAH) of $< 12^\circ$ without describing how they do the measurement and what droplet is used (water or blood?). Nevertheless, this value does not fit with water rolling-off angle of 3.6° . The difference is larger than expected for a superhydrophobic surface, I should expect the CAH would around 5° or smaller.

Reviewer #3:

Remarks to the Author:

The manuscript by Li et al. describes superhydrophobic carbon nanofiber (CNF) and polymer composites as potential hemostatic materials. They developed these CNF composites with different polymers including PDMS and found that CNFs promote quick fibrin growth and cause rapid clotting. Also, due to their superhydrophobic nature, CNF can severely limit blood wetting to prevent blood loss, and drastically reduce bacteria attachment, thus facilitating wound healing. Furthermore, minimal contact between clot and the superhydrophobic CNF surface yields an unforced clot detachment after clot shrinkage. Overall, this work suggests a novel and impactful strategy for designing more efficient hemostatic patch materials compared to currently used cotton

gauze. The experiments are well performed and the claims are supported by the data. However, advantages of this approach over previously developed hemostatic materials is not clear. For example, Dowling et al developed a hydrophibically modified chitosan as a hemostatic agent that can achieve hemostasis within a minute. It would be nice if authors can compare their material to such 1-2 best previously developed materials, preferably in in vivo experiment or at least in vitro experiment. Additionally, the authors have not discussed any limitations of this system. In fact, the discussion section is very small and mainly repeats the results and conclusions. The authors should add limitations of their system, future avenues and also previously developed approaches. Finally, the manuscript has multiple grammatical mistakes, which must be corrected. Overall, this reviewer suggests acceptance of this manuscript for publication after addressing these concerns.

Response to Reviewers' Comments

We would like to thank the reviewers for the constructive remarks and input that help us further improve the manuscript. Each comment from the reviewers has been carefully considered and addressed. We conducted additional experiments, including *in vivo* skin compatibility tests, a comparison between our material and 3 representative commercial hemostatic products on patch peeling force after clotting, and a repeated hemoglobin absorbance test.

Detailed point-by-point responses to each comment are listed in the following, with corresponding changes in the revised manuscript highlighted in blue for easy tracking.

Responses to Reviewer 1

Comment 1 from Reviewer 1: The authors mentioned that “active clotting agent pose a safety threat (side-effects), of micro-thrombosis”, however, how about the materials used to form PDMS? PDMS requires cross-linker for polymerization, which also has threat to human body.

Response to Comment 1: We thank reviewer on pointing out the safety concern over PDMS. After long enough curing time at an elevated temperature, the cross-linker will be exhaustively cross-linked with the pre-polymer (the recommended curing time for Sylgard 184 PDMS is 20 min at 120°C or 2 days at 25°C^{1,2}); also, according to the technical data sheet, “no hazardous polymerization will occur”¹, and there is “no solvents or cure byproducts” during polymerization². Furthermore, medical grade PDMS has clinically proven biocompatibility and nontoxicity, with wide applications in contact lens³, implanted medical devices such as the cochlear implants⁴ and drug delivery⁵.

To further clarify this point, we have performed *in vivo* skin compatibility test by patching our CNF/PDMS material on rat skin for 12 hours (hair shaved; tested on 3 rats; see Fig. R1 below). Compared with the skin in contact with the pristine gauze (control), the skin in contact with our CNF/PDMS gauze appeared normal and no itching or erythema was observed after 12 hours (Fig. R1c; please note that 12 hours significantly exceeds the expected time that the gauge is intended to stay in contact with the wounded skin). Considering the clinically proven safety of PDMS and the skin compatibility test of our material, the hemostatic material developed in this study would be safe for hemostatic applications. Information on skin compatibility test is provided on Page 16 and Page 27 in the revised manuscript, and Supplementary Fig. 17.

Fig. R1. *In vivo* skin compatibility test. (a) Schematic illustration of the prepared sample. (b) Attaching our CNF/PDMS gauze onto rat skin. (c) After gauze removal, the skin was gently cleaned with 70% v/v ethanol to examine any skin reaction; after 12 hours, the skin in contact with our CNF/PDMS gauze appeared normal, and no itching or erythema was observed.

Comment 2 from Reviewer 1: “Similar research is reported by Jun-Yon Park et al., “Biodegradable polycaprolactone nanofibers with beta-chitosan and calcium carbonate produce a hemostatic effect”(Polymer 123(2017)194-, however, the authors do not refer the paper. The authors should compare their results with previous report at least, and should mention their intensive points or advantages.”

Response to Comment 2: We have added this paper as Ref. 15 in the revised manuscript. Below we reproduce verbatim the relevant part in the revised manuscript.

“hydrophilic hemostatic material could also be prepared by spray coating β -chitosan on the porous nanofibre mat¹⁵, and the hydrophilic β -chitosan coating could increase blood wettability and thus enhance clotting.” (Page 3, Line 36-38)

This paper is also used as a reference for our method of measuring the clotting performance (the haemoglobin absorbance test, Page 11, Line 207 and Line 209).

It is important to clarify that although chitosan-based materials described in Ref. 15 offer hemostatic action, their chemistry and surface texture lacks the additional functionalities which are enabled by our material. Our material provides distinctive and novel functions of (1) blood-repellence, due to its superhydrophobic nature, (2) facile removal after wound healing, due to low adhesion with the skin, and (3) antibacterial action, owing to the low adhesion of bacteria on our material. In contrary, the porosity and hydrophilic property of the aforementioned chitosan-based material induce a strong blood absorbing capability (the interaction with bacteria was not reported). This blood absorbing capability would lead to strong wound adhesion after clot solidification/contraction, which is also a major limitation of existing hemostatic products.

In response to reviewer's comments, we have highlighted the advantages of our material over previous reports in the revised manuscript (Page 17-18, Line 354-371).

Comment 3 from Reviewer 1: "Please clarify how the authors obtained "Hemoglobin absorbance" in Fig.3."

Response to Comment 3: Description on the measurement of "hemoglobin absorbance" has been elaborated in the revised manuscript. In short, hemoglobin released from un-clotted red blood cells was measured to reflect the degree of clotting. Specifically, after blood was allowed to clot on different gauze for a given time, coagulation was terminated by adding 10 ml DI water. Un-clotted red blood cells would hemolyze and release haemoglobin into water. Optical absorbance of the resulting solution at different clotting time t , $HA(t)$, was measured by the spectrophotometer at 540 nm, following published protocols⁶⁻⁸. In Fig. 3d, we use the relative hemoglobin absorbance $RHA(t)$ at different clotting time t for comparison. $RHA(t)$, the relative hemoglobin absorbance at time t , was calculated as $RHA(t) = HA(t)/HA(0)$; $HA(0)$, the haemoglobin absorbance at $t = 0$ min (for 20 μ l blood in 10 ml DI water), was used as reference. As hemoglobin came from un-clotted red blood cells, a lower hemoglobin absorbance value or a smaller $RHA(t)$ would mean faster clotting (or fewer un-clotted red blood cells).

In response to this comment, detailed description on "the measurement of hemoglobin absorbance in Fig. 3" is provided in the revised manuscript, as shown in the following:

d The relative hemoglobin absorbance $RHA(t)$ plot, showing the fast clotting performance of the CNF gauze; since the experiment was performed in a petri dish, clotting in the petri dish without any gauze was used as control; the absolute hemoglobin absorbance at clotting time t , $HA(t)$, was measured by the spectrometer at 540 nm ($n = 3$); $HA(0)$, the hemoglobin absorbance at $t = 0$ min, was used as the reference; the relative hemoglobin absorbance at clotting time t , $RHA(t)$ equals to $HA(t)/HA(0)$; as the hemoglobin came from un-clotted red blood cells, a lower hemoglobin absorbance value would mean faster clotting." (Under Fig. 3, Page 10)

"Non-clotted red blood cells (free blood cells not trapped in the clot) would hemolyze and release hemoglobin into water^{15, 24}. Optical absorbance of the resulting hemoglobin solution at different clotting time t , $HA(t)$, measured by the spectrophotometer at 540 nm, would represent the amount of hemoglobin from un-clotted red blood cells; $HA(0)$, the absolute hemoglobin absorbance of 20 μ l blood in 10 ml DI water at $t = 0$ min, was used as reference; $RHA(t)$, the relative hemoglobin absorbance at clotting time t , was calculated as $HA(t)/HA(0)$ (averaged over 3 repetitions, $n = 3$; using the same batch of blood)⁸. Since the hemoglobin came from un-clotted red blood cells, a smaller $RHA(t)$ would mean faster clotting^{2,8,24}." (Page 23, Line 505-512).

Responses to Reviewer 2

Comment 1 from Reviewer 2: “I think no doctor will attach excessive bleeding wound with gauze only, antibacterial medicines will be applied on the wound. So superhydrophobic feature is nice for gauze materials, it’s helpful on holding bleeding, but it is not major improvement.”

Response to Comment 1: While we agree that some antibacterial medicine may be used in wound treatment, we believe doctors are expecting more advanced materials like ours, if applicable, to prevent hemorrhage and avoid complications (secondary infection/bleeding) in treatments. For instance, in military combat, excessive blood loss from a wound is responsible for almost 90% of deaths⁹; life-saving treatment therefore requires immediate blood loss prevention that is offered by our material. Our material could also reduce the risk of secondary infection/bleeding, by preventing wound tearing. Secondary infection related to wound tearing in the gauge change process affects a large number of patients (some secondary infection may even be life-threatening)¹⁰. Besides the anti-bacterial capability (owing to low adhesion of bacteria on our surface, see Fig. 3b in the manuscript), we are pleased to present new data on comparing the peeling force of our material with commercial products (Fig. R2, or Supplementary Fig. 15); our patch can reduce the peeling force by 24-52 times, despite these commercial products being labeled as low-adherence or pain-free removal. Our material therefore helps reduce the risk of secondary infection by preventing wound tearing with an unprecedentedly low peeling force.

Fig. R2. Comparison of the clot peeling tension of our material with 3 representative commercial hemostatic products. (a) Commercial hemostatic products: #1 is the “minimal pain on removal” dressing from Smith & Nephew; #2 is the “pain-free removal” pad from 3M; #3 is the “non-adherent pad for easy removal” from Guardian; the substrate material is 3M Nexcare. (b) Procedures used to measure the clot peeling tension. (c) Summary of the clot peeling tension and the contact angle for different samples. Contact angle was measured with 10 μ l DI water ($n = 5$) on different hemostatic materials; peeling tension was calculated as the maximum peeling force divided by the width (repeated for 3 times; $n = 3$). Clot peeling tension on our material is about 24-52 times smaller than commercial products.

Also, the ability of our material to stop bleeding quickly and minimize blood loss due to its superhydrophobic nature must be emphasized. Our material can prevent blood loss before clotting is complete; this will be very useful for large wounds (such as in the military combat⁹). Further, we are pleased to demonstrate new data showing that our material can cause blood to clot faster than the chitosan gauze as well (Fig. R3, or Supplementary Fig. 14). Description of this test is presented on Page 22, Line 487-495, and Page 23, Line 521-524, in the revised manuscript.

Fig. R3 The relative hemoglobin absorbance $RHA(t)$ plot, comparing the clotting performance between our CNF gauze and the chitosan gauze. A smaller $RHA(t)$ means faster clotting. It shows that our CNF gauze can have a better clotting performance than the chitosan gauze.

Further, an ideal hemostatic patch should have this sequence of functionalities: 1) preventing blood loss, 2) sealing an open wound, and 3) facilitating the patch/dressing removal without secondary bleeding. Most of the existing materials¹¹⁻¹⁴ would be efficient only in the second part (sealing the wound). In comparison, our material has demonstrated a combination of several distinctive and novel features (preventing blood loss before clotting is finished, fast clotting, low adhesion after clotting, and anti-bacteria); we believe these proven advantages will make our material design approach a major improvement in the area of hemostatic materials.

Comment 2 from Reviewer 2: The biocompatibility and biotoxicity of the superhydrophobic gauze material need to be investigated. To be noted PTFE is non-degradable, is there antigenic reaction if PTFE particles come inside the blood vessel? And what about PDMS and CNF ?

Response to Comment 2: We thank reviewer's comment on biocompatibility and bio-toxicity of our material. Regarding PTFE, it is indeed non-degradable, but it is considered biocompatible¹⁵; as for biodegradability, it is not a major concern in the external hemostatic application. Furthermore, PTFE undergoes a thermal annealing step at 400°C during the fabrication of our material; PTFE particles are molten and form a continuous polymeric network. Therefore, no PTFE particles are present in the final material, and no PTFE particles will come inside the blood vessel. Nevertheless, we use PDMS as the polymeric matrix for immobilizing CNFs in *in vitro* and *in vivo* studies. This was because the cotton gauze substrate cannot withstand the high annealing temperature for PTFE (400°C), while PDMS can cross-link at a much lower temperature. As for PDMS, medical grade PDMS has been widely used in contact lens³, implanted medical devices such as the cochlear implants⁴, and drug delivery⁵. Given its clinically proven biocompatibility and nontoxicity, the use of PDMS in our material system (for temporary hemostatic application) should be safe.

As for the CNF, multi-wall CNFs (similar to those used in our study) or CNF & PDMS composites were reported to be non-toxic for cell growth^{16,17}, despite the controversies over the bio-compatibility of CNF in the literature¹⁶⁻²⁰. For instance, the vitality of fibroblast L929 cells, cultivated on a 3D multi-wall CNF network for 7 days, was essentially the same as that on a control surface¹⁶; and skin fibroblast cells cultivated on a CNF & PDMS nano-composite surface for one week had a cell viability exceeding 95%¹⁷. Regarding the risk of free CNFs entering the blood vessel, we note that CNFs are immobilized onto the substrate by PDMS and not free on the surface; to remove any loosely attached CNFs, we exposed prepared CNF/PDMS gauze samples to compressed N₂ gas from a spray gun (nozzle diameter: 0.8 mm; pressure: 430 kPa; sample to nozzle distance: ~15 cm) before test; further, blood is flowing outwards and clotting quickly in contact with the CNF surface, making it unlikely for CNFs to enter the blood vessel.

To remove doubt, we further performed *in vivo* skin compatibility test to evaluate the safety of our material (see Fig. R1). Our CNF gauze was patched onto rat skin (with hair shaved). After 12 hours, the skin in contact with CNF gauze appeared normal, and no itching or erythema was observed (Fig. R1c; description of this test is presented on Page 16 and Page 27 in the revised manuscript, and also in Supplementary Fig. 17).

Therefore, based on the published studies on cell non-toxicity of CNF or CNF & PDMS composite^{16,17} and our *in vivo* skin compatibility test (Fig. R1), our material would be safe for hemostatic application.

Comment 3 from Reviewer 2: The authors spent substantial words to describe the fibrin formation when blood contacts their material, but haemoglobin absorbance-time plot doesn't show a substantial improvement on the effect of coagulation. Please explain why?

Response to Comment 3: Regarding this comment, we would like to point out that the blood contact area on our superhydrophobic CNF gauze was much smaller than that on the hydrophilic normal gauze. The uncoated normal gauze was hydrophilic and absorbed blood quickly, leading to a large blood contact area; our superhydrophobic CNF gauze repelled blood and its blood contact area was only $14.2 \pm 0.7\%$ ($n = 3$) of that on the uncoated pristine gauze (Supplementary Fig. 10b). Therefore, considering blood contact area, the hemoglobin absorbance per area on our CNF gauze would be about 7 times smaller than that of the normal gauze, showing a substantial improvement on the effect of coagulation.

To further demonstrate the improvement in coagulation, we compared the clotting performance of our material with the gauze coated with an active clotting agent, chitosan. Our material was shown to have a better clotting performance than the chitosan gauze (a smaller hemoglobin absorbance in Fig. R3). Description on the preparation of the chitosan gauze is provided in detail in "Methods" (Page 22, Line 487-495), and the results (Fig. R3) are shown in Supplementary Fig. 14.

Comment 4 from Reviewer 2: The authors claim the sample has a low actual contact angle hysteresis (CAH) of $< 12^\circ$ without describing how they do the measurement and what droplet is used (water or blood?). Nevertheless, this value does not fit with water rolling-off angle of 3.6° . The difference is larger than expected for a superhydrophobic surface. I should expect the CAH would around 5° or smaller.

Response to Comment 4: We thank reviewer for pointing out this flaw in our last manuscript. Contact angle hysteresis (CAH) was measured using the tilting method²¹, by placing a 20 μl droplet on the sample and tilting the sample until roll-off of the droplet. CAH was measured for 5 different liquids, including water, blood, plasma, and blood/plasma with anti-thrombin.

In response to the inquiry on "how the CAH was measured, and what droplet is used", we have added the following description in the revised manuscript:

"d. Dynamic contact angles for water droplets, and blood or PPP droplets with and without anti-thrombin on the superhydrophobic CNF/PTFE Ti surface; contact angle hysteresis $CAH = \theta_a - \theta_r$, and nominal contact angle hysteresis $CAH_{nom} = \theta_a - \theta_{r,nom}$; ...droplet volume: 20 μl ;" (Page 8, Line 131-134)

"Dynamic contact angles, including advancing contact angle θ_a , receding contact angle θ_r , contact angle hysteresis CAH ($CAH = \theta_a - \theta_r$), nominal receding angle $\theta_{r,nom}$, nominal contact angle hysteresis

CAH_{nom} (CAH_{nom} = $\theta_a - \theta_{r, \text{nom}}$), and roll-off angle θ were measured using the tilting method⁶⁰, by placing a 20 μl droplet (for water, blood and PPP with or without anti-thrombin) on the sample surface and tilting the sample till droplet roll-off.” (Page 20, Line 429-434)

As for the comment “this value does not fit with water rolling-off angle”, we have provided roll-off-angle (RA) and contact angle hysteresis (CAH) measurements for 5 different liquids, which are formally reported in Fig. 2d, and summarized below for the reviewer’s convenience.

- (1) Water: RA = $1.1 \pm 0.3^\circ$ and CAH = $0.8 \pm 0.5^\circ$;
- (2) Blood: RA = $28.1 \pm 11.6^\circ$ and CAH = $11.5 \pm 1.7^\circ$;
- (3) PPP: RA = $15.0 \pm 6.6^\circ$ and CAH = $13.7 \pm 4.3^\circ$;
- (4) Blood with anti-thrombin: RA = $4.3 \pm 1.4^\circ$ and CAH = $21.7 \pm 6.7^\circ$;
- (5) PPP with anti-thrombin: RA = $2.6 \pm 1.3^\circ$ and CAH = $13.3 \pm 1.2^\circ$.

We note that, for water, its RA is similar to CAH, just as the reviewer expected. For blood and PPP, RA is larger than CAH; this is due to the long fibrin fibers generated at the receding side, which pull the droplet from sliding off (see Fig. 1b and d, and Fig. 2c in the manuscript).

However, for blood and PPP with anti-thrombin, the RA is smaller than CAH (RA < 10°, while CAH > 10°), which would be due attributed to the effect of anti-thrombin. We re-measured RA and CAH for blood and PPP droplets, and confirmed that “RA is smaller than CAH for blood/PPP with anti-thrombin”. Here, we provide Fig. R4, showing the blood/PPP droplets with anti-thrombin in 4 typical tests (at the moment before quick roll-off), to demonstrate the difference between RA and CAH for blood/PPP with anti-thrombin. In Fig. R4, we see an uncharacteristically small receding contact angle (or pinning of the droplet at the receding side). Due to the well-known dose effect, drugs like anti-thrombin do not completely eliminate fibrin generation, and some micro-fibrin fibers could still exist at the liquid-solid interface. In the rolling test, as the droplet attempts to detach from our CNF surface at the receding edge, these micro-fibrin fibers would be elongated (as illustrated in Fig. R4e), leading to a small receding contact angle; but the effect of these micro fibrin fibers is limited, and does not significantly increase blood adhesion on our surface, thus the roll-off angle is still small.

In the end, we want to highlight that the objective of this sliding test with anti-thrombin (as shown in Fig. 2b in the manuscript, and the Fig. 4R below) is only to prove that long fibers generating at the receding side are fibrin fibers. Silencing fibrin formation can eliminate these fibers and allow droplets roll off at a smaller RA. A brief explanation on the difference between RA and CAH has been provided in the revised manuscript (Page 8, Line 140-145).

Fig. R4 Advancing angle and receding angle for (a-b) PPP and (c-d) blood droplets (20 μ l) with anti-thrombin in 4 typical tests before the moment of quick droplet roll-off, showing a relatively small receding contact angle and the pinning of the droplet at the receding side. (e) Illustration of the micro-fibrin fibers which would still exist due to the dose effect of anti-thrombin; these micro-fibrin fibers would lead to a smaller receding contact angle and thus a large contact angle hysteresis.

Responses to Reviewer 3

Comment 1 from Reviewer 3: “Advantages of this approach over previously developed hemostatic materials is not clear. For example, Dowling et al developed a hydrophibically modified chitosan as a hemostatic agent that can achieve hemostasis within a minute. It would be nice if authors can compare their material to such 1-2 best previously developed materials, preferably in in vivo experiment or at least in vitro experiment.”

Response to Comment 1: We thank for reviewer’s suggestion on further highlighting the advantages of our material. Our material/approach has the distinctive and novel feature of “stop bleeding first, reinforce clotting subsequently, and unforced easy removal in the end”, which offers the following advantages: 1) stop bleeding immediately upon application; 2) achieve rapid clotting by the nano-engineered surface structure, without the help of active clotting agents/chemicals; 3) reduce the clotting peeling force to an unprecedentedly low level. Compared with commercial hemostatic products, clot peeling tension of our material was about 24-52 times smaller (Fig. R2c, or Supplementary Fig. 15c).

In order to compare the performance of our material with existing hemostatic materials, we have performed the following work.

(1) We compared the clotting speed of our material with the gauze coated with active clotting agent chitosan (results shown in Fig. R3). Our CNF/PDMS material is shown to have a better clotting

performance than the chitosan gauze.

(2) As for the hydrophobically modified chitosan (hm-chitosan) suggested by the reviewer, we performed a literature search and found 3 papers on hm-chitosan's *in vivo* performance¹²⁻¹⁴. Using the *in vivo* data on hm-chitosan from these studies, we have made a quantitative comparison in Fig. R5 (or Supplementary Fig. 18) between our material and the hm-chitosan regarding 4 important hemostatic performance measurements,

Compared with hm-chitosan, our material stops bleeding instantaneously upon application, without blood loss; but it takes time for the hm-chitosan to form a blood gel to stop bleeding, with inevitable bleeding (the blood loss was reported to be about 29.3 to 48.9 ml/kg¹² or about 10-30 ml¹³). Regarding the clotting time, our CNF gauze can form a clot within 3 minutes based on our *in vivo* observation (Fig. 5b in the manuscript), while the clotting time for hm-chitosan is affected by its application dose and manner (< 1 min when used in liquid phase¹⁴, > 2 min when used as bandage¹⁴, or “within minutes” when used as a sprayable foam¹³). Thus, hm-chitosan can be better at faster clotting. As for the clot peeling, our material naturally detached with a very small peeling force, and did not adhere on the wound, while the hm-chitosan bandage was reported to adhere on the wound due to the anchoring of hm-chitosan onto tissue cells.¹⁴

Fig. R5 Comparison between our material and the hydrophobically modified chitosan (hm-chitosan). Hm-chitosan¹²⁻¹⁴ is slightly faster in forming a clot to seal the wound; our material is superior in reducing the bleeding time, reducing the blood loss, and decreasing the peeling force.

Thus, as summarized in Fig. R5, hm-chitosan could be faster in generating a blood gel/clot, but our material is superior in reducing the bleeding time (bleeding time is virtually zero), reducing the blood loss (nearly no blood loss), and decreasing the peeling force.

(3) We compared the clot peeling force of our material *in vitro* with 3 representative hemostatic

products that are marketed to be “pain-free” or “non-adherent” on removal (see Fig. R2a). Compared with these commercial products, clot peeling tension of our material was 24-52 times smaller (see Fig. R2c, or Supplementary Fig. 15c). Therefore, owing to the utilized superhydrophobic technology, our material design strategy has reduced the clotting peeling force of hemostatic materials to an unprecedentedly low level.

Based on the discussion above, we have highlighted the advantages of our material, and also compared our material with the hydrophobically modified chitosan and 3 commercial products in the revised manuscript, as shown in the following:

“Different from existing hemostatic products/materials, our material works in a novel and distinctive manner of “stop bleeding first, reinforce clotting subsequently, and enable unforced natural detachment in the end”, which offers the following advantages. First, the superhydrophobic feature helps stop bleeding immediately upon application. Bleeding therefore ceases before the formation of a strong clot. This can be life-saving for hemorrhage such as in severe accidents and military combats⁵³, as it takes time for clotting to occur even with the most effective hemostatic materials such as QuickClotTM and HemogripTM 7,54. Second, rapid clotting is achieved by the nano-engineered surface structure, without the help of active clotting agents/chemicals. Our nano-engineered fibrous structure promotes quick growth of fibrin network to seal the wound. Clotting agents are routinely used in hemostatic products, such as the hydrophobically modified chitosan⁵⁴⁻⁵⁶, which may provide shorter clotting time than our material. However, our material brings with it a distinct strategy of hastening coagulation using a nano-structuring approach also assuring facile removal, a critical feature to avoid secondary bleeding (a quantitative comparison between our material and the hydrophobically modified chitosan⁵⁴⁻⁵⁶ is provided in Supplementary Fig. 18). Our material yields a reduction of the clotpeeling force, by a good order of magnitude, down to an unprecedentedly low level. Compared with commercial hemostatic products, the clot peeling tension of our material is about 24-52 times smaller (Supplementary Fig. 15c). This natural and facile clot detachment feature is unique to our superhydrophobic hemostatic material, and has not been reported before according to our knowledge.” (Page 17-18, Line 354-371)

Comment 2 from Reviewer 3: “Additionally, the authors have not discussed any limitations of this system. In fact, the discussion section is very small and mainly repeats the results and conclusions. The authors should add limitations of their system, future avenues and also previously developed approaches.”

Response to Comment 2: We have enriched the discussion by highlighting the advantages of our material/strategy. Discussion on previously developed approaches has been as shown above in “Response to Comment 1” (Page 17-18, Line 354-371). Discussion on limitations of our current system and future avenues are also added in the revised manuscript, as shown below.

“In the current study, we used non-biodegradable carbon nano-fibers to design the nano-engineered

surface. Biodegradable nano-fibers could be developed for internal use. Also, clot-promoting performance can be further enhanced by performing a parametric study of the surface topographical features and chemical composition. With these endeavors, the material design strategy developed here will lead to a more efficient hemostatic product for clinical use in the near future.” (Page 18, Line 377-382).

Comment 3 from Reviewer 3: “Finally, the manuscript has multiple grammatical mistakes, which must be corrected. Overall, this reviewer suggests acceptance of this manuscript for publication after addressing these concerns.”

Response to Comment 3: We have made a careful proof-reading, and corrected the grammatical mistakes. There should be no language mistake in the revised manuscript.

References

- 1 SYLGARD™184 Silicone Elastomer Kit.
<http://louisville.edu/micronano/files/documents/safety-data-sheets-sds/PDMSBase.pdf>
- 2 Information about Dow Corning® brand Silicone encapsulants.
<http://bdml.stanford.edu/twiki/pub/Rise/PDMSProceSS/PDMSdatasheet.pdf>
- 3 Nicolson, P. C. & Vogt, J. Soft contact lens polymers: an evolution. *Biomaterials* **22**, 3273-3283 2001.
- 4 Goldfinger, Y., Natan, M., Sukenik, C. N., Banin, E. & Kronenberg, J. Biofilm prevention on cochlear implants. *Cochlear implants international* **15**, 173-178, 2014.
- 5 Aliyar, H. & Schalau, G. Recent developments in silicones for topical and transdermal drug delivery. *Therapeutic delivery* **6**, 827-839, 2015.
- 6 Kumar, P. T. *et al.* Flexible and microporous chitosan hydrogel/nano ZnO composite bandages for wound dressing: in vitro and in vivo evaluation. *ACS applied materials & interfaces* **4**, 2618-2629, 2012.
- 7 Quan, K. *et al.* Black hemostatic sponge based on facile prepared cross-linked graphene. *Colloids and surfaces. B, Biointerfaces* **132**, 27-33, 2015.
- 8 Park, J.-Y., Kyung, K.-H., Tsukada, K., Kim, S.-H. & Shiratori, S. Biodegradable polycaprolactone nanofibres with β -chitosan and calcium carbonate produce a hemostatic effect. *Polymer* **123**, 194-202, 2017.
- 9 Kelly, J. F. *et al.* Injury severity and causes of death from Operation Iraqi Freedom and Operation Enduring Freedom: 2003–2004 versus 2006. *Journal of Trauma and Acute Care Surgery* **64**, S21-S27, 2008.
- 10 National Strategy for the Monitoring, Prevention and Control of Healthcare-Associated Infections (NOSO Strategy). 2019.
- 11 Trabattoni, D. *et al.* A new kaolin-based haemostatic bandage compared with manual compression for bleeding control after percutaneous coronary procedures. *European radiology* **21**, 1687-1691 2011.

- 12 Dowling, M. B. *et al.* Hydrophobically-modified chitosan foam: description and hemostatic efficacy. *The Journal of surgical research* **193**, 316-323, 2015.
- 13 Dowling, M. B. *et al.* Sprayable Foams Based on an Amphiphilic Biopolymer for Control of Hemorrhage Without Compression. *ACS Biomaterials Science & Engineering* **1**, 440-447, 2015.
- 14 Dowling, M. B. *et al.* A self-assembling hydrophobically modified chitosan capable of reversible hemostatic action. *Biomaterials* **32**, 3351-3357, 2011.
- 15 Risbud, M. V., Hambir, S., Jog, J. & Bhonde, R. Biocompatibility assessment of polytetrafluoroethylene/wollastonite composites using endothelial cells and macrophages. *Journal of Biomaterials Science, Polymer Edition* **12**, 1177-1189, 2001.
- 16 Correa-Duarte, M. A. *et al.* Fabrication and biocompatibility of carbon nanotube-based 3D networks as scaffolds for cell seeding and growth. *Nano Letters* **4**, 2233-2236, 2004.
- 17 Ha-Chul, J. *et al.* CNT/PDMS composite flexible dry electrodes for long-term ECG monitoring. *IEEE Transactions on Biomedical Engineering* **59**, 1472-1479, 2012.
- 18 Schrand, A. M., Dai, L., Schlager, J. J., Hussain, S. M. & Osawa, E. Differential biocompatibility of carbon nanotubes and nanodiamonds. *Diamond and Related Materials* **16**, 2118-2123, 2007.
- 19 Foldvari, M. & Bagonluri, M. Carbon nanotubes as functional excipients for nanomedicines: II. Drug delivery and biocompatibility issues. *Nanomedicine: Nanotechnology, Biology and Medicine* **4**, 183-200, 2008.
- 20 Chłopek, J. *et al.* In vitro studies of carbon nanotubes biocompatibility. *Carbon* **44**, 1106-1111, 2006.
- 21 Eral, H. B., 't Mannetje, D. J. C. M. & Oh, J. M. Contact angle hysteresis: a review of fundamentals and applications. *Colloid and Polymer Science* **291**, 247-260, 2012.

Reviewers' Comments:

Reviewer #1:

Remarks to the Author:

The authors revised the manuscript according to the Reviewer's comments.

Reviewer #2:

Remarks to the Author:

The manuscript has greatly improved and is for me suitable for publication.

Reviewer #3:

Remarks to the Author:

Authors have addressed all the comments. The manuscript has significantly improved and can be considered for publication.